# Health-related quality of life and economic burden of childhood pneumonia in China: a multiregion study

Junyang Gao,[1] Jingzhi Fan,[1] Huijun Zhou,[2] Mark Jit,[3] Pei Wang  [1]

JG and JF contributed equally.

[1]School of Public Health, Fudan University, Shanghai, Shanghai, China
[2]Business School, University of Shanghai for Science and Technology, Shanghai, Shanghai, China
[3]Department of Infectious Disease Epidemiology, London School of Hygiene and Tropical Medicine, London, UK

**Correspondence to**
Dr Pei Wang; wang_p@fudan.edu.cn

## ABSTRACT

**Objective** To systematically investigate the health-related quality of life (HRQOL) and economic burden of children with pneumonia in different regions of China.

**Study design** The study recruited a series of children under 5 years hospitalised for pneumonia in Shanghai, Zhengzhou and Kunming from January to October 2019. Health utility was assessed using the proxy version of EQ-5D-Y by interviewing patients' guardians face to face. The assessment was administered twice at patients' admission and discharge. Cost incurred for receiving the hospitalisation was collected. Multiple linear regression and quantile regression were used to explore factors of EQ-5D-Y Health Utility Score (HUS) and costs, respectively.

**Results** A total of 501 paediatric patients with a median age (IQR) of 1.5 (0.83–2.71) years were included in the analysis. The mean HUS (SD) of the patients was 0.78 (0.18) at admission, and increased to 0.96 (0.10) at discharge. Some patients (14.2%) still felt worried, sad or unhappy after hospitalisation. The mean hospitalisation cost and total cost were RMB5859 (€773) and RMB6439, respectively. The HUS was lower and the economic burden was heavier for the children in Zhengzhou. Apart from region, type of work, insurance status and hospital days were also related to the baseline HUS or HUS increment after treatment; insurance status, Visual Analogue Scale score at discharge, guardians' employment and hospitalisation days were associated with the costs.

**Conclusion** The children with pneumonia have poor baseline HRQOL, and many of them still have psychological well being problems after treatment. The economic burden varied significantly across regions and is heavy for the patients' families in less developed areas (ie, Zhengzhou and Kunming).

## INTRODUCTION

Pneumonia is a significant type of lower respiratory tract infection affecting children worldwide,[1] with serious impact on both physical and psychological aspects of health, such as coughing, breathlessness and irritability.[2] It is the major single cause of death in children, causing approximately 1 million deaths annually, or 15% of an estimated 6.3 million deaths in children aged under 5 years.[3] Studies have also shown that pneumonia is one of the top

### WHAT IS ALREADY KNOWN ON THIS TOPIC

⇒ Many studies have found that children with pneumonia had poorer health-related quality of life (HRQOL), especially in social and psychological functioning dimensions; pneumonia also placed a heavy economic burden on the children's family.

### WHAT THIS STUDY ADDS

⇒ The study provided Health Utility Score (HUS) for Chinese children with pneumonia. The study also compared HRQOL and economic burden among different regions in China.

### HOW THIS STUDY MIGHT AFFECT RESEARCH, PRACTICE OR POLICY

⇒ We found that many paediatric patients still had psychological well being problems after treatment, suggesting that their psychological status deserves more attention during and after clinical care. We also identified significant regional differences in HRQOL and economic burden, as well as other influencing factors.

five causes of death in children under 5 years in China, accounting for 12.4% of deaths.[4 5] Pneumonia causes not only physical impairment, but also psychological, debilitating and social adjustment problems to the affected child. Hence, health-related quality of life (HRQOL), a comprehensive health outcome measure is required to holistically reflect the disease influence. Moreover, the HRQOL information could also be translated into Health Utility Score (HUS) reflecting the value of HRQOL for use in economic evaluation if it is measured by utility instruments such as EQ-5D.

International and domestic studies have assessed HRQOL of paediatric patients with pneumonia using HRQOL instruments such as the Children's Quality of Life Scale (PedsQL 4.0)[6 7] and the Generic Quality of Life Inventory-74.[8] They found that the paediatric patients had poorer HRQOL, especially

in social and psychological functioning dimensions. Studies in Indonesia and Thailand have also reported HUS of the patients using the proxy version of EQ-5D.[9 10] However, no studies have yet provided available data on HUS for the Chinese paediatric patients.

In China, some kinds of the costs incurred from pneumonia treatment, such as tests, anti-infective drugs, and bed charges, are not fully reimbursed according to Chinese healthcare system. Patients thus have to pay for the excess costs themselves. As a result, pneumonia also places a heavy financial burden on the children's family. Several studies have assessed its economic burden in childhood respiratory infections and pneumonia in China.[11 12] Wang et al investigated the average length of stay (LOS) in hospital and hospitalisation cost of 8334 children with acute respiratory disease under 5 years in Gansu province from 2015 to 2018. They found that the LOS was 6.6 days and the expenditure was RMB5613 exceeding 30% of the per capita disposable income of the province in 2018. Similarly, Wang et al conducted a retrospective study of 86 children with pneumonia under 5 years in a community in Shanghai during 2012. The study indicated the average total cost including outpatient cost, hospitalisation cost, out-of-pocket drug cost, traffic cost, lost work cost was RMB4017, which accounted for 10% of the per capita disposable income of Shanghai in 2012. However, those studies were based on the patients in a single place, which may not be representative of the broader Chinese population and thus lead to limited extrapolation.

Hence, this study aimed to systematically investigate the HRQOL and economic burden of children with pneumonia from three different regions in China with divergent economic development levels.

## METHODS
### Study design and patients
This study measured HRQOL of hospitalised pneumonia children at admission and discharge. Various costs associated with hospitalisation were collected and aggregated. The study was conducted from January to October 2019 in three provincial capitals of sampled provinces representing different socioeconomics statuses. The cities were Shanghai (eastern region), Zhengzhou (central region) and Kunming (western region), representing high, medium and low status, respectively. In each city, a general or special (eg, children's hospital) tertiary hospital was selected as the sampling hospital.

The inclusion criteria for the study were (1) younger than 5 years old, (2) clinical diagnosis of pneumonia and (3) no concomitant diseases. All the paediatric patients hospitalised in the studying sites at the time of study were assessed for the eligibility by trained interviewers. Once a patient was eligible, his/her parents/guardians were invited to participate in the study.

The consenting parents/guardians were interviewed face-to-face twice. The first interview was conducted within the first 2 days of patients' admission. A questionnaire was used to assess the child's HRQOL and to collect information on the child's (ie, gender, age and medical insurance) and his/her guardian's sociodemographic characteristics (ie, gender, age, education level, marriage status, employment and monthly income level, race, domicile, religion, relationship with the child). The second interview was performed when the child was discharged from the hospital, and assessed the child's HRQOL again, as well as the total hospitalisation costs including any household expenses, insurance payments (if any), lost work time and the LOS.

### Health-related quality-of-life measurement
The EuroQol Five-Dimensional Questionnaire, Youth Version (EQ-5D-Y, Y-5L) is used as the HRQOL measure in the study. It is the youth version of the widely used EQ-5D that is more aligned with children's perceptions and understanding. It has five dimensions each with five response options: walking about, looking after myself, doing usual activity, having pain or discomfort, feeling worried sad or unhappy.

The five response options of each dimension of Y-5L are no problems (level 1), a little bit of problems (level 2), some problems (level 3), a lot of problems (level 4), extreme problems (level 5).[13] As a result, it defines a total of 3125 ($5^5$) health states by combing the responses of each dimension. A HUS can be assigned to each health state using a utility value set. Since the utility value set for the Y-5L is not available currently, the study used the Chinese value set of the EQ-5D-5L to calculate the HUS for the Y-5L health states.[14] In addition, Y-5L uses a Visual Analogue Scale (VAS) with a score of 100 at the top for 'the best imaginable health' and 0 at the bottom for 'the worst imaginable health'.

Given that the children aged 0–5 years were cared by their parents, it was assumed that the dimension 'looking after myself' is irrelevant. For the children under 18 months who were unable to stand, the dimension 'walking about' is assumed irrelevant.[15 16]

### Economic burden of pneumonia
The economic burden of pneumonia was estimated by calculating direct medial cost (ie, hospitalisation cost including out-of-pocket cost, insurance payments if covered by insurance), non-medical cost (ie, traffic cost), as well as indirect cost. Indirect cost was derived by estimating the loss of productivity of guardian using human capital approach. That is, the loss of work hour multiplies hourly wages based on annual average income in China in 2021 (8 hours per work day and 250 work days per year). All cost data were adjusted to 2021 Chinese CNY using the latest published CPI index in China.

### Statistical analysis
Descriptive statistical analyses were adopted to depict sociodemographic characteristics of the patients and their guardians, distributions of Y-5L dimension responses, Y-5L HUS and VAS score, as well as economic burden

 Gao J, et al. BMJ Paediatrics Open 2023;7:e002031. doi:10.1136/bmjpo-2023-002031

**Table 1** Sociodemographic characteristics of children with pneumonia and their guardians among the three regions

| Variables | Overall N=501 | Shanghai N=173 | Zhengzhou N=120 | Kunming N=208 | P value* |
|---|---|---|---|---|---|
| Characteristics of the child | | | | | |
| Gender | | | | | <0.01 |
| Boys | 258 (51.5) | 72 (41.6) | 63 (52.5) | 123 (59.1) | |
| Girls | 243 (48.5) | 101 (58.4) | 57 (47.5) | 85 (40.9) | |
| Age(year) | | | | | <0.01 |
| Median | 1.50 | 1.00 | 2.50 | 1.17 | |
| IQR | 0.83–2.71 | 0.92–2.00 | 2.08–3.50 | 0.40–2.33 | |
| Medical Insurance | | | | | <0.01 |
| Yes | 185 (36.9) | 91 (52.6) | 41 (34.2) | 53 (25.5) | |
| None | 243 (48.5) | 80 (46.2) | 79 (65.8) | 84 (40.4) | |
| Unknown | 71 (14.2) | – | – | 71 (34.1) | |
| Guardianship characteristics | | | | | |
| Age (year) | | | | | 0.04 |
| Median | 31.00 | 31.00 | 32.50 | 31.00 | |
| IQR | 28.00–35.00 | 28.00–37.00 | 30.00–35.00 | 27.00–34.00 | |
| Gender | | | | | <0.01 |
| Male | 209 (41.7) | 57 (32.9) | 42 (35.0) | 110 (52.9) | |
| Female | 292 (58.3) | 116 (67.1) | 78 (65.0) | 98 (47.1) | |
| Relationship | | | | | <0.01 |
| Father | 189 (37.7) | 35 (20.2) | 42 (35.0) | 112 (53.8) | |
| Mother | 270 (53.9) | 105 (60.7) | 78 (65.0) | 87 (41.8) | |
| Other† | 42 (8.4) | 33 (19.1) | – | 9 (4.3) | |
| Ethnic | | | | | <0.01 |
| Han | 454 (90.6) | 171 (98.8) | 118 (98.3) | 165 (79.3) | |
| Others | 47 (9.4) | 2 (1.2) | 2 (1.7) | 43 (20.7) | |
| Religion | | | | | 0.01 |
| None | 485 (96.8) | 170 (98.3) | 118 (98.3) | 197 (94.7) | |
| Yes or refused to answer | 13 (2.6) | 3 (1.7) | 2 (1.6) | 11 (5.3) | |
| Marriage status | | | | | 0.25 |
| Married | 491 (98.0) | 170 (98.3) | 120 (100.0) | 201 (96.6) | |
| Singles and others | 10 (2.0) | 3 (1.8) | – | 7 (3.4) | |
| Domicile | | | | | <0.01 |
| City | 241 (48.1) | 102 (59.0) | 95 (79.2) | 44 (21.2) | |
| Town | 97 (19.4) | 43 (24.9) | 20 (16.7) | 34 (16.3) | |
| Rural | 161 (32.1) | 27 (15.6) | 5 (4.2) | 129 (62.0) | |
| Education | | | | | <0.01 |
| Elementary school and below | 28 (5.6) | 11 (6.3) | – | 17 (8.2) | |
| Junior high school | 104 (20.8) | 14 (8.1) | 2 (1.7) | 88 (42.3) | |
| High school | 88 (17.6) | 30 (17.3) | 15 (12.5) | 43 (20.7) | |
| College | 68 (13.6) | 23 (13.3) | 25 (20.8) | 20 (9.6) | |
| University and above | 213 (42.5) | 95 (54.9) | 78 (75.0) | 40 (19.2) | |
| Employment | | | | | <0.01 |
| Enterprise/self-employed | 210 (41.9) | 77 (44.5) | 83 (69.2) | 50 (24.0) | |
| Civil servants/public institution | 83 (16.6) | 20 (11.6) | 24 (20.0) | 39 (18.8) | |

 

**Table 1** Continued

| Variables | Overall N=501 | Shanghai N=173 | Zhengzhou N=120 | Kunming N=208 | P value* |
|---|---|---|---|---|---|
| Full-time home/housewife | 73 (14.6) | 37 (21.4) | 5 (4.2) | 31 (14.9) | |
| Retirement | 21 (4.2) | 18 (10.4) | – | 3 (1.4) | |
| Farming | 50 (10.0) | 11 (6.4) | 3 (2.5) | 36 (17.3) | |
| Temporary workers | 39 (7.8) | 1 (0.6) | 4 (3.3) | 34 (16.3) | |
| Unemployment | 21 (4.2) | 7 (4.1) | 1 (0.8) | 13 (6.2) | |
| Other or refused to answer | 3 (0.6) | 1 (0.6) | – | 2 (1.0) | |
| Family monthly income (CNY) | | | | | <0.01 |
| <CNY5000 (€660) | 83 (16.6) | – | 3 (2.5) | 80 (38.5) | |
| CNY5000–CNY10 000 | 97 (19.4) | – | 20 (16.7) | 77 (37.0) | |
| CNY10 000–CNY20 000 | 76 (15.2) | 9 (5.2) | 47 (39.2) | 20 (9.6) | |
| CNY20 000–CNY30 000 | 75 (15.0) | 37 (21.4) | 34 (28.3) | 4 (1.9) | |
| >CNY30 000 | 117 (23.4) | 94 (54.3) | 16 (13.3) | 7 (3.3) | |
| Unknown or refused to answer | 52 (10.4) | 32 (18.5) | – | 20 (9.6) | |

*Comparison of population characteristics among the three regions. Categorical variables and continuous variables were analysed using $\chi^2$ test and ANOVA test, respectively.
†This refers to a guardian such as a non-parental relative of the affected child.
ANOVA, analysis of variance.

data. Categorical variables were presented using number and percentages; continuous variables were described using mean, SD and range; and age is described using median and IQR. $\chi^2$ test, median test or analysis of variance test were used to compare the characteristics of the patients/guardians, HRQOL and economic burden of the patient among the three cities, and HRQOL between admission and discharge whenever appropriate. Comparison of HUS, VAS score and cost using the Tukey's post hoc test[17] between samples were further conducted to determine which pairwise groups were different.

Multiple linear regression was used to analyse the factors influencing the HUS of children with pneumonia. HUS at baseline and its difference between baseline and discharge were used as the two dependent variables. According to previous studies,[18 19] hospitalisation days and demographic characteristics including region, gender, age, insurance status of the children; and education level, employment of the guardians may have a significant effect on HUS. These factors were adopted as the independent variables in the two models (hospital days was adopted only in the model for HUS difference).

For the economic burden, quantile regression was used to analyse the impact factors. Referring to the previous studies,[20 21] three quantile points of 10%, 50% and 90% of hospitalisation cost and total cost (hospitalisation cost, transportation cost and cost for productivity loss) were selected separately for the quantile regression analysis. The two costs were log-transformed to approximate normal distributions as the two dependent variables. Based on the evidence of prior studies,[11 12 22 23] gender, age, hospital days, insurance status and disease prognosis

may have a significant impact on costs and adopted as independent variables in the two models. Also, VAS score reflecting the disease prognosis, region and employment of guardians were also adopted in the models.

SPSS V.23.0 and SPSS V.26.0 were used for all the analysis, and the threshold for significant differences was a p value less than 0.05.

## RESULTS
### Characteristics of study subjects
Table 1 shows the sociodemographic characteristics of children with pneumonia and their guardians. A total of 501 patients were included in analysis: 173, 120 and 208 in Shanghai, Zhengzhou and Kunming, respectively. The median age (IQR) of the patients was 1.50 (0.83–2.71) years, and the proportions of boys and girls were close (51.5% vs 48.5%). A relatively high percentage of the children were without medical insurance (48.5%). More than half of the guardians were female (53.9%) with a median age (IQR) of 31.00 (28.00–35.00) years, and the majority of them were parents of the children (91.6%). They were mostly Han Chinese (90.6%), non-religious (96.8%), married (98.0%), from urban area (67.5%), university-educated or above (42.5%), enterprise/self-employed (41.9%). Most of them (57.8%) had family monthly income over CNY10 000 (€1319). The characteristics of the patients and their guardians differed among the three regions except for marital status (p=0.25).

**Table 2** EQ-5D-Y health utility and EQ-VAS scores of children with pneumonia in the three locations

| | Total | Shanghai (1) | Zhengzhou (2) | Kunming (3) | Post hoc analysis* | P value† |
|---|---|---|---|---|---|---|
| | Admission: N=394 Discharge: N=372 | Admission: N=66 Discharge: N=44 | Admission: N=120 Discharge: N=120 | Admission: N=208 Discharge: N=208 | | |
| **Utility admission** | | | | | | |
| Mean (SD) | 0.78 (0.18) | 0.84 (0.07) | 0.66 (0.21) | 0.82 (0.15)) | (1)>(2); (3)>(2) | <0.001 |
| Range | 0.04–1.00 | 0.63–1.00 | 0.04–0.95 | 0.25–1.00 | | |
| **VAS admission** | | | | | | |
| Mean (SD) | 62.9 (16.6) | 63.5 (14.2) | 58.3 (15.5) | 65.4 (17.4) | (3)>(2) | <0.01 |
| Range | 15–100 | 40–95 | 15–84 | 20–100 | | |
| **Utility discharge** | | | | | | |
| Mean (SD) | 0.96 (0.10) | 0.97 (0.04) | 0.90 (0.15) | 0.99 (0.04) | (1)>(2); (3)>(2) | <0.001 |
| Range | 0.27–1.00 | 0.85–1.00 | 0.27–1.00 | 0.67–1.00 | | |
| **VAS discharge** | | | | | | |
| Mean (SD) | 89.2 (13.0) | 77.2 (9.5) | 88.9 (10.7) | 91.9 (13.5) | (2)>(1); (3)>(1) | <0.01 |
| Range | 40–100 | 60–95 | 47–100 | 40–100 | | |
| **Utility difference** | | | | | | |
| Mean (SD) | 0.19 (0.16) | 0.15 (0.06) | 0.24 (0.21) | 0.17 (0.14) | (1)<(2); (2)>(3) | <0.001 |
| Range | 0.05–1.04 | 0.04–0.32 | 0–1.04 | 0.05–0.75 | | |
| **VAS difference** | | | | | | |
| Mean (SD) | 27.1 (13.5) | 20.1 (7.7) | 30.6 (14.3) | 26.5 (13.4) | (2)>(1); (2)>(3) | <0.01 |
| Range | 0–74 | 5–40 | 10–74 | 0–60 | | |

*Comparison using the Tukey post hoc test.
†Calculated using ANOVA test.
ANOVA, analysis of variance; EuroQol, Five-Dimensional Questionnaire, Youth Version; VAS, Visual Analogue Scale.

### Health-related quality-of-life of the children

Among the 501 patients, 394 patients responded to the EQ-5D-Y scale at admission, and 372 patients responded again at discharge. The percentage of reporting problems in the Y-5L dimensions at admission and discharge is shown in online supplemental file 1. At admission, the children had more problems in having pain or discomfort (85.8%) and feeling sad, worried, or unhappy (83.0%) dimensions; and the children in Zhengzhou had the highest prevalence of problems in each dimension. At discharge, the children's health status was significantly improved and no cases had very severe problems (level 5) in all the dimensions, while a relatively high proportion of problems was still observed in feeling worried, sad or unhappy dimension (19.0%). Similarly, the children in Zhengzhou tended to have more problems.

Table 2 reports the Y-5L HUS and VAS score of children with pneumonia in the three locations. The mean HUS of the patients at admission was 0.78, and the patients in Shanghai and Zhengzhou reported the highest (0.84) and lowest HUS (0.66), respectively. The HUS at discharge was significantly increased with the mean value being 0.96, and the children from Kunming and Zhengzhou had higher (0.99) or lower (0.90) HUS. The results of multiple comparisons showed significant difference in HUS between Shanghai and Zhengzhou, Kunming and

Zhengzhou at both admission and discharge, while the difference between Shanghai and Kunming was insignificant. The difference in HUS at admission and discharge for children in Zhengzhou was significantly greater than the difference in the other two regions. The mean VAS score of children in the three sites at both admission and discharge was also significantly different (62.9 vs 89.2, p<0.05), with the children in Kunming having the highest score at both admission (65.4) and discharge (91.9), those in Zhengzhou and Shanghai having the lowest score at admission (58.3) and at discharge (77.2), respectively

### Economic burden of pneumonia

The economic burden manifested by hospitalisation cost and total cost summing of hospitalisation cost, transportation cost and indirect cost are shown in table 3. The mean hospitalisation cost of the three regions was RMB5859 and the children in Zhengzhou (RMB8667) had much higher cost than that in Shanghai (RMB3100) and Kunming (RMB6522). The mean transportation cost was RMB110 and the children in Kunming (RMB220) had the highest cost followed by the children in Shanghai (RMB4) and Zhengzhou (RMB73). The average hospitalisation days was 8.0 days: the days were 10.6 days, 8.2 days and 4.8 days for the children in Kunming, Shanghai and

**Table 3** Economic burden of disease in children with pneumonia

| | Overall | Shanghai | Zhengzhou | Kunming | Post hoc comparison* | P value† |
|---|---|---|---|---|---|---|
| | N=501 | (1) | (2) | (3) | | |
| Traffic cost | | | | | (1) < (3); (2) < (3) | <0.01 |
| Mean (SD) | 110.0 (270.1) | 4.3 (9.8) | 72.9 (138.2) | 219.9 (378.6) | | |
| Range | 0–3514 | 0–60 | 0–780 | 0–3514 | | |
| Out-of-pocket cost | | | | | (1)<(2)<(3) | <0.01 |
| Mean (SD) | 3816.1 (3981.9) | 2036.6 (1077.7) | 3813.8 (1990.3) | 5229.1 (5473.3) | | |
| Range | 471–60420 | 471–8236 | 1350–10416 | 504–60420 | | |
| Hospitalisation cost | | | | | (1)<(2); (1)<(3); (2)>(3) | <0.01 |
| Mean (SD) | 5859.2 (5486.5) | 3099.7 (1474.2) | 8666.6 (3965.2) | 6521.5 (7098.9) | | |
| Range | 302–80560 | 987–8261 | 3114–18998 | 302–80560 | | |
| Loss of work time (hour) | | | | | – | <0.01 |
| Mean (SD) | 37.9 (81.1) | 6.5 (3.9) | 33.1 (15.8) | 65.9 (121.0) | | |
| Range | 0–1200 | 0–10 | 9–80 | 0–1200 | | |
| Indirect cost‡ | | | | | (1)<(3); (2)<(3) | <0.01 |
| Mean (SD) | 465.8 (967.9) | 187.4 (171.7) | 443.5 (215.6) | 710.2 (1444.5) | | |
| Range | 0–15399.6 | 0–390.1 | 107.2–1072.4 | 0–15399.6 | | |
| Hospitalisation days | | | | | – | <0.01 |
| Mean (SD) | 8.0 (6.6) | 4.8 (1.9) | 8.2 (3.0) | 10.6 (9.1) | | |
| Range | 0–50 | 1–11 | 5–15 | 0–50 | | |
| Total cost§ | | | | | (1)<(3)<(2) | <0.01 |
| Mean (SD) | 6439.4 (5773.0) | 3292.5 (1511.2) | 9183.0 (4085.7) | 7463.6 (7437.9) | | |
| Range | 918.1–80560.5 | 986.9–8701.5 | 3370.1–19346.8 | 918.1–80560.5 | | |

*Comparison using the Tukey post hoc test.
†Calculated using ANOVA test.
‡Indirect cost: loss of work hour multiples income per hour that converted from annual average income in China 2021 (8 hours per work day and 250 work days per year).
§Total cost includes hospitalisation cost, traffic cost and indirect cost.
ANOVA, analysis of variance.

**Table 4** Multiple regression analysis of baseline HUS and HUS difference between baseline and follow-up

| Variables | Baseline HUS | | HUS difference (discharge admission) | |
|---|---|---|---|---|
| | Coefficient (95% CI)* | P value* | Coefficient (95% CI)* | P value* |
| Region (Shanghai†) | | | | |
| Zhengzhou | −0.17 (−0.22 to 0.12) | <0.01 | 0.06 (0.01 to 0.12) | 0.02 |
| Kunming | 0.04 (−0.01 to 0.1) | 0.12 | −0.06 (−0.12 to 0) | 0.06 |
| Gender | −0.02 (−0.05 to 0.02) | 0.32 | 0.01 (−0.02 to 0.04) | 0.48 |
| Age | 0.002 (−0.01 to 0.02) | 0.72 | −0.003 (−0.02 to 0.01) | 0.63 |
| Insurance status | −0.001 (−0.001 to 0) | 0.01 | 0.001 (0 to 0.001) | 0.06 |
| Education (elementary school and below†) | | | | |
| Junior high school | −0.001 (−0.09 to 0.09) | 0.98 | 0.00 (−0.09 to 0.09) | 1.00 |
| High school | −0.03 (−0.09 to 0.03) | 0.29 | 0.02 (−0.05 to 0.08) | 0.64 |
| College | −0.03 (−0.09 to 0.02) | 0.20 | 0.04 (−0.02 to 0.09) | 0.16 |
| University and above | 0.02 (−0.03 to 0.07) | 0.43 | −0.04 (−0.09 to 0.01) | 0.11 |
| Employment status (enterprise/self-employed†) | | | | |
| Civil servants/public Institution | −0.02 (−0.07 to 0.02) | 0.30 | 0.03 (−0.02 to 0.07) | 0.21 |
| Full-time home/housewife | 0.03 (−0.03 to 0.09) | 0.39 | −0.03 (−0.09 to 0.03) | 0.33 |
| Retirement | 0.06 (−0.05 to 0.16) | 0.27 | −0.03 (−0.15 to 0.09) | 0.60 |
| Farming | −0.03 (−0.09 to 0.03) | 0.32 | 0.02 (−0.04 to 0.08) | 0.50 |
| Temporary workers | −0.08 (−0.14 to 0.02) | 0.01 | 0.05 (−0.01 to 0.11) | 0.08 |
| Unemployment | −0.07 (−0.15 to 0.02) | 0.13 | 0.04 (−0.05 to 0.13) | 0.37 |
| Hospitalisation days | – | – | 0.01 (0.002 to 0.01) | <0.01 |

*Calculated using multiple linear regression.
†Reference category.
HUS, Health Utility Score.

Zhengzhou, respectively. Correspondingly, the average time of loss work of guardians in Kunming (65.9 hours) was substantially higher than that in Shanghai (6.5 hours) and Zhengzhou (33.1 hours). This also resulted in the highest indirect cost in Kunming (RMB710), followed by Zhengzhou (RMB444) and Shanghai (RMB187). For the total cost, the average amount was RMB 6439, and the patients in Zhengzhou had the highest amount (RMB 9183), followed by Kunming (RMB 7464), and Shanghai (RMB3293).

## Regression analyses

Table 4 presents the significant coefficients generated from the multiple linear regressions. Region, insurance status and type of work were significantly associated with baseline HUS (table 4). That is, the children in Zhengzhou, without insurance, and their guardians who were temporary workers had lower baseline HUS. Region and hospitalisation days were significantly correlated with the difference in HUS: the children in Zhengzhou and with longer hospital days were more likely to have higher difference in HUS (table 4).

The results of quantile regression analysis showed that region, insurance status, hospital days, VAS score at discharge and employment of guardians were significantly associated with both hospitalisation and total costs

(table 5). Specifically, region had a significant effect at almost all quartiles, and the children in Zhengzhou and Kunming had higher costs. In addition, the degree of the effect was higher at the lower quartile than at the higher quartile for children in Zhengzhou. The two costs were significantly lower for the children without insurance and with shorter hospitalisation days at the middle and high quartiles, and the effects of hospitalisation days were stronger in the higher quartile. For children with higher VAS score at discharge, the two costs were significantly higher at the low and middle quantiles. At the low quantile, the two costs were significantly lower for children whose guardians were civil servant or public institution worker, and total cost was also lower for the children with retired working guardians. For children with guardians who were temporary workers, the two costs were significantly lower at almost all quartiles.

## DISCUSSION

We found that the children with pneumonia had inferior health status both physically and psychologically; and the disease economic burden varied greatly among regions. Moreover, we also identified several important factors influencing the HUS and economic burden. Therefore,

**Table 5** Quantile regression analysis of total cost and hospitalisation cost

| Variables | Hospitalisation cost | | | Total cost† | | |
|---|---|---|---|---|---|---|
| | q=0.1 | q=0.5 | q=0.9 | q=0.1 | q=0.5 | q=0.9 |
| Region (Shanghai‡) | | | | | | |
| Zhengzhou | 1.06** | 0.91** | 0.71** | 0.96** | 0.86** | 0.74** |
| Kunming | 0.35 | 0.56** | 0.53** | 0.62** | 0.64** | 0.66** |
| Gender | −0.03 | −0.07 | −0.01 | −0.03 | −0.07 | −0.01 |
| Age | −0.03 | 0.002 | −0.01 | 0.002 | 0.01 | −0.002 |
| Insurance status | 0.003 | −0.003* | −0.004* | 0.001 | −0.004** | −0.004** |
| VAS discharge | 0.02** | 0.01* | 0.001 | 0.01** | 0.01* | 0.000 |
| Employment status (Enterprise/self-employed‡) | | | | | | |
| Civil servants/public institution | −0.31* | −0.07 | −0.04 | −0.26** | −0.08 | −0.04 |
| Full-time home/housewife | −0.04 | 0.08 | 0.07 | −0.12 | 0.03 | 0.04 |
| Retirement | −0.40 | −0.22 | 0.14 | −0.56* | −0.36 | 0.04 |
| Farming | −0.13 | −0.05 | −0.11 | −0.03 | −0.05 | 0.04 |
| Temporary workers | −0.54* | −0.23* | −0.26* | −0.32* | −0.16 | −0.27* |
| Unemployment | 0.13 | 0.07 | 0.13 | −0.08 | 0.05 | 0.03 |
| Hospitalisation days | 0.01 | 0.02** | 0.08** | 0.01 | 0.02** | 0.07** |

*p<0.05, **p<0.01.
†Total cost includes hospitalisation cost, traffic cost and indirect cost.
‡Reference category.
VAS, Visual Analogue Scale.

this study deepens the understanding of HRQOL and economic burden of the children in China.

At admission, the majority of children had health problems especially in having pain or discomfort, feeling worried, sad or unhappy dimensions as well as low HUS and VAS score. Previous studies have also reported similar findings that the paediatric patients with pneumonia had poor HRQOL at admission, especially in social and psychological functioning dimensions.[6 7] When pneumonia occurred, the children had symptoms such as fever, cough, sore throat and breathlessness,[24] which may bring physical discomfort to the children, affect their emotions and lead to psychological problems. We further found that although their health improved a lot at discharge, many of them were still feeling worried, sad or unhappy at discharge. Thus, their psychological condition should deserve more attention in the postdischarge period.

Among the three regions, the children in Zhengzhou had worse baseline HUS, which was probably due to that they were mainly from urban area containing more risk factors that may contribute to the development of pneumonia in childhood.[25] The children without insurance also reported poorer baseline HUS. Some studies revealed that individuals who have health insurance reported a higher quality of life than those without health insurance.[26 27] In addition, the children also had poorer baseline HUS when their guardians were temporary workers. An Israeli study showed that both parental unemployment and workday loss affected the HRQOL of children with pneumonia as well.[28] Temporary work means

unstable income, making it difficult to provide good care to the children. The children in Zhengzhou had larger HUS difference. Since the patients were discharged from hospitals because their health was recovered and their HUS at discharge was close to full health, the identified association could be attributed to the HUS at baseline. That is, the children in Zhengzhou had worse baseline HUS. In addition, children with longer hospital days had higher HUS difference, probably because longer treatment time corresponds to better treatment effect.

We found that the average hospitalisation and total costs were comparable to the previous results for the Chinese children under 5 years with pneumonia[11 12]: the average hospitalisation cost was RMB5771 in Gansu and the average total cost was RMB4642 in Shanghai after being adjusted for 2021 CNY. The costs were also similar to the costs of other common childhood infectious diseases in China. According to 2021 CNY, the average hospitalisation costs for children in Shenyang with scarlet fever and hand, foot and mouth disease were RMB4979 and RMB5050, respectively[29]; the average total cost for inpatient cases with chickenpox in school-age children was RMB3312 in Shenzhen.[30] On the other hand, we identified the disease economic burden varied greatly among the regions: the total cost was calculated to be 1.9% of the local per capita GDP in Shanghai, 9.6% in Zhengzhou and 8.7% in Kunming, respectively. The finding indicated that the disease places severer economic burden on the children's families in less economically developed areas. The government thus could take necessary activities (eg,

increasing the reimbursement rate of medical insurance) to alleviate the burden of the families.

The children in Zhengzhou had higher hospitalisation cost. This may be due to their worse baseline health status, requiring more medical resources and thus higher costs. For the transportation and indirect costs, the patients in Kunming had higher amount. Kunming is the capital city of Yunnan province, which had a relatively low level of socioeconomic development and a lack of medical resources compared with other regions of China. Hence, the patients in other cities of the province have to concentrate in Kunming, leading to the higher costs.

The children without medical insurance had significantly lower hospitalisation and total costs. A previous study had indicated that relying solely on out-of-pocket payment provided obstacles for treatment access, skewing treatment seeking towards those affordable,[23] leading to lower costs. The two costs were also significantly associated with hospitalisation days, and similar findings had been reported in a previous study that shortening the number of ineffective hospitalisation days helped to reduce the financial burden.[22] In the study, we found that children with higher VAS score at discharge had higher hospitalisation and total costs. This may be because achieving better health required more consumption of medical services. We also found that the two costs were significantly higher for the children whose guardians were enterprise or self-employed workers. This was probably because they may have better economic conditions and were willing to spend more for their children to get better treatment.

This study has several limitations which should be noted. First, the study was conducted in three areas in China, which may not be representative of the entire country or other regions with different socioeconomic statuses. Also, only paediatric patients who were hospitalised at the time of the study were included; while those who were managed on an outpatient basis or treated at primary care clinics were not included. In addition, there were missing values in the responses to the EQ-5D-Y dimensions for the patients in Shanghai. These may have limited the representativeness of the study population and affected the generalisability of our findings. Second, due to the study design and resources, we did not enrol a control group of children without pneumonia for comparison. Thus, it may be difficult to attribute the observed results exclusively to pneumonia. Third, the Y-5L was designed mainly for children over 6 years of age, and the reliability and validity of its proxy version need to be further tested when it is applied to the guardians of children under 5 years. Also, the Chinese Y-5L value set has not been available, so the Chinese EQ-5D-5L value set was used to calculate the HUS instead. This introduces some uncertainty in the accuracy of the HRQOL measurements. Fourth, we did not collect certain clinical characteristics including disease severity and treatment modality, thus their influence on HRQOL and economic burden cannot be assessed. Fifth, although we excluded the pneumonia patients with other acute diseases, there may be undiagnosed comorbidities during data collection, which may affect HRQOL scores. Last but not least, the study lacked long-term follow-up, thus it was not possible to know the trajectory of HRQOL and economic burden over time.

## CONCLUSION

The paediatric patients with pneumonia mainly suffered from pain, discomfort and felt sad, worried or unhappy at admission, and many of them still had psychological well being problems after treatment. Region, insurance status, hospitalisation days and employment of guardians were the four factors influencing their HRQOL. The economic burden of the disease varied significantly across regions and was heavy for the patients' families in less developed areas. Insurance status of the patients, employment status of their guardians, VAS score at discharge and hospitalisation days were also associated with the burden.

**Contributors** JG: methodology, data analysis, manuscript writing (original draft). JF: data analysis, manuscript writing (original draft and revision). PW: conceptualisation, methodology, critical manuscript revision, supervision and project administration. All authors critically revised the manuscript and approved the final version of manuscript. PW had full access to all the data in the study and take responsibility for the integrity of the data and the accuracy of the data analysis.

**Funding** This research was funded by the National Institute for Health Research (NIHR) (GHR 16/137/09) using UK aid from the UK Government to support global health research.

**Disclaimer** The views expressed in this publication are those of the author(s) and not necessarily those of the NIHR or the UK government.

**Competing interests** None declared.

**Patient and public involvement** Patients and/or the public were not involved in the design, or conduct, or reporting, or dissemination plans of this research.

**Patient consent for publication** Consent obtained from parent(s)/guardian(s).

**Ethics approval** This study involves human participants and this study was conducted according to the guidelines of the Declaration of Helsinki and approved by the Institutional Review Board of School of Public Health, Fudan University (IRB00002408). Participants gave informed consent to participate in the study before taking part.

**Provenance and peer review** Not commissioned; externally peer reviewed.

**Data availability statement** Data are available on reasonable request.

**ORCID iD**
Pei Wang http://orcid.org/0000-0003-3661-3944

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
