## [Reviewer comments · BMJ Paediatrics Open]

This paper was submitted to a another journal from BMJ but declined for publication following peer review. The authors addressed the reviewers' comments and submitted the revised paper to BMJ Paediatrics Open. The paper was subsequently accepted for publication at BMJ Paediatrics Open.

ARTICLE DETAILS

TITLE (PROVISIONAL)	Health-related quality of life and economic burden of childhood pneumonia in China: a multi-region study
AUTHORS	Gao, Junyang Fan, Jingzhi Zhou, Huijun Jit, Mark Wang, Pei

VERSION 1 - REVIEW

REVIEWER	Dr. Surendra Gupta Childrens Medical Center of Fresno, Pediatrics
REVIEW RETURNED	17-May-2023

GENERAL COMMENTS	Overall, the study aims to investigate the health-related quality of life (HRQOL) and economic burden of children with pneumonia in three regions of China. It provides valuable information on the impact of pneumonia on children's well-being and the financial costs associated with the disease. Scope of improvement: To improve the study's introduction, consider the following suggestions: Provide a clear and concise research objective: Begin by explicitly stating the main objective of the study. For example: "The objective of this study was to systematically investigate the health-related quality of life (HRQOL) and economic burden of children with pneumonia in different regions of China." Start with a general context: Begin the introduction by providing a brief overview of pneumonia as a significant lower respiratory tract infection affecting children worldwide, emphasizing its impact on both physical and psychological well-being. Provide global and local statistics: Incorporate recent global statistics on pneumonia cases, emphasizing the disease burden in developing countries, such as the five Asian and African countries mentioned in the original text. Additionally, highlight specific statistics about pneumonia in China, particularly its prevalence among children under 5 years and its ranking among the top causes of death. Address the gap in knowledge: Mention that previous studies have assessed HRQOL of pediatric patients with pneumonia in other countries using specific instruments, such as the Children's Quality of Life Scale (PQL 4.0) and the Generic Quality of Life Inventory-74
---

(GQOLI-74). However, state that there is currently no available data on health utility scores (HUS) for Chinese patients with pneumonia. Discuss the economic burden of pneumonia: Briefly mention previous studies that have assessed the economic burden of pneumonia in China, as described in the original text. However, highlight the limitation of these studies, which focused on patients in single locations and thus may not be representative of the broader population.

Clearly state the research aim: Conclude the introduction by restating the aim of the study, which is to investigate the HRQOL and economic burden of children with pneumonia in three regions of China with different levels of economic development.

By implementing these suggestions, the introduction will provide a clearer and more focused overview of the study, highlighting the significance of the research and addressing the existing knowledge gap.

Limitations: It is important to mention following limitations in discussion when interpreting the findings of the study and to acknowledge the potential impact they may have on the generalizability and validity of the results.

Limited generalizability: The study was conducted in three provincial capitals in China, which may not be representative of the entire country or other regions with different socioeconomic statuses. Therefore, the findings of the study may not be applicable to the broader population of children with pneumonia in China.

Selection bias: The study only included pediatric patients who were hospitalized at the time of the study, which may introduce selection bias. Children with less severe cases of pneumonia who were managed on an outpatient basis or treated at primary care clinics were not included in the study. This could limit the representativeness of the study population and affect the generalizability of the findings.

Lack of control group: The study did not include a control group of children without pneumonia for comparison. Having a control group would have allowed for a better understanding of the specific impact of pneumonia on HRQOL and economic burden, as compared to children without the disease. Without a control group, it is difficult to attribute the observed outcomes solely to pneumonia.

Self-reporting bias: The study relied on self-reporting of HRQOL by the children's parents/guardians, which introduces the possibility of reporting bias. Parents/guardians may have subjective perceptions or may not accurately assess the HRQOL of their children. Objective measures or clinical assessments could have provided a more comprehensive and accurate evaluation of HRQOL.

Lack of long-term follow-up: The study only measured HRQOL and economic burden at admission and discharge from the hospital.

Long-term follow-up data would have provided valuable insights into the trajectory of HRQOL and economic burden over time and allowed for a more comprehensive assessment of the impact of pneumonia on children.

Furthermore, the study primarily focuses on the economic burden and health-related quality of life (HRQOL) of children with pneumonia, but it does not consider other potential factors that could impact these outcomes. For example, the study does not explore the impact of comorbidities or the severity of pneumonia on HRQOL and economic burden. The inclusion criteria for the study should include clear definitions or criteria for clinical diagnosis of pneumonia and no concomitant diseases. These factors could significantly influence the

	results and should be considered for a more comprehensive understanding of the topic. Finally, the study uses the EQ-5D-Y instrument to measure HRQOL, which is a widely used tool. However, the study acknowledges that the utility value set for the EQ-5D-Y is not available, and therefore, it uses the Chinese value set of the EQ-5D-5L to calculate health utility scores. This introduces some uncertainty in the accuracy of the HRQOL measurements.
--	---

REVIEWER	Dr. Peter Flom Peter Flom Consulting
REVIEW RETURNED	23-May-2023

GENERAL COMMENTS	I confine my remarks to statistical and methodological aspects of this paper. Unfortunately, there were some fairly big issues, some of which may not be remediable. General: For cost, it is unlikely that OLS regression will be appropriate. First, the residuals are very unlikely to be normally distributed with constant variance (which is an assumption). Second, I think it's like that interest is not exclusively in the mean cost, but in the higher quantiles (and maybe the lower ones, as well). That is, the researchers probably would be interested in which kids have very high or very low cost. Therefore, I suggest using quantile regression, which makes no assumptions about the residuals and allows investigation of the quantiles. p. 3 The fact that those 5 countries are 54% of pneumonia cases is not too surprising, since they are 46% of the world's population. p. 5 The paragraph starting "The five response options " is very confusing. Did the parents use a VAS or a set of ordinal responses? p. 6 Please say how the regression models were built -- that is, how were variables selected? The fact that the data are not normally distributed is not relevant, OLS regression makes assumptions about the residuals, but I agree that the residuals are likely to be non-normal. However, rather than take log of costs (which is going to be kind of hard to interpret) I suggest quantile regression (see above). p. 7 For child age it would be better to use median and IQR rather than mean and SD. After all, kids can't be less than 0 years old. (In the table, you do give the range, which helps, but IQR would also be good). p. 8 Don't just give the parameters for significant predictors, give them for all predictors (see above about how the models were built). Table 2 - there should not be any blanks in this table. Also, the column %ages should add to 100, but they do not. E.g. the total for the first column and first set of values is about 74%. If the rest are missing, then that is a significant problem that needs to be dealt with (but it might not be possible to deal with it well). This problem
--

	happens in many cases. Also, assuming that very young children have "no problem" in walking about is problematic. I agree that they wouldn't be walking about, even if completely healthy, but giving them all a 0 distorts the amount of problems they are having. I think multiple imputation could be used for the very young kids, but I'm not sure about the other missing data. Table 5 - see above.
--	---

VERSION 1 – AUTHOR RESPONSE

Responses to Reviewer#1's comments

Overall, the study aims to investigate the health-related quality of life (HRQOL) and economic burden of children with pneumonia in three regions of China. It provides valuable information on the impact of pneumonia on children's well-being and the financial costs associated with the disease.

Scope of improvement: To improve the study's introduction, consider the following suggestions:

Provide a clear and concise research objective: Begin by explicitly stating the main objective of the study. For example: "The objective of this study was to systematically investigate the health-related quality of life (HRQOL) and economic burden of children with pneumonia in different regions of China."

Our response: Thanks for this comment. In this revision, we have refined the description of research objective. (Page 5, line 11)

Original : Hence, this study aimed to systematically investigate the HRQOL and economic burden of the children with pneumonia from three regions with divergent economic development levels in China.

Revised: Hence, this study aimed to systematically investigate the HRQOL and economic burden of children with pneumonia from three different regions in China with different economic development levels.

Start with a general context: Begin the introduction by providing a brief overview of pneumonia as a significant lower respiratory tract infection affecting children worldwide, emphasizing its impact on both physical and psychological well-being.

Our response: Thanks for this comment. In this revision, we have modified the beginning of the introduction. (Page 4, line 2)

Original: Pneumonia is a significant type of lower respiratory tract infection that is caused by different pathogens such as bacteria and viruses(1).

Revised: Pneumonia is a significant type of lower respiratory tract infection affecting children worldwide(1), with serious impact on both physical and psychological aspects of health, such as coughing, breathlessness and irritability(2).

Provide global and local statistics: Incorporate recent global statistics on pneumonia cases, emphasizing the disease burden in developing countries, such as the five Asian and African countries mentioned in the original text. Additionally, highlight specific statistics about pneumonia in China, particularly its prevalence among children under 5 years and its ranking among the top causes of death.

Our response: Thanks for this comment. In this revision, we have adopted the recent global statistics on pneumonia cases and highlighted the statistics in China. However, there is currently no available data on the prevalence among children under 5 years in China. (Page 4, line 4)

Original: The disease burden is more serious in developing countries. In 2019, the number of pneumonia and severe pneumonia cases in five Asian and African countries (i.e., India, Nigeria, Indonesia, Pakistan and China) accounted for 54% of the total cases globally(2); studies have also shown that pneumonia is one of the top 5 causes of death in children under 5 years in China(3, 4).

Revised: It is the major single cause of death in children, causing approximately 1 million deaths annually, or 15% of an estimated 6.3 million deaths in children aged under 5 years(3). Studies have also shown that pneumonia is one of the top 5 causes of death in children under 5 years in China, accounting for 12.4% of deaths(4, 5).

Address the gap in knowledge: Mention that previous studies have assessed HRQOL of pediatric patients with pneumonia in other countries using specific instruments, such as the Children's Quality of Life Scale (PQL 4.0) and the Generic Quality of Life Inventory-74 (GQOLI-74). However, state that there is currently no available data on health utility scores (HUS) for Chinese patients with pneumonia.

Our response: Thanks for this comment. In this revision, we have highlighted the gap in data on HUS for children with pneumonia in China. (Page 4, line 20)

Original: while no studies have yet provided HUS for the Chinese patients.

Revised: However, no studies have yet provided available data on HUS for the Chinese pediatric patients.

Discuss the economic burden of pneumonia: Briefly mention previous studies that have assessed the economic burden of pneumonia in China, as described in the original text. However, highlight the

limitation of these studies, which focused on patients in single locations and thus may not be representative of the broader population.

Our response: Thanks for this comment. In this revision, we have highlighted the limitation of these studies. (Page 5, line 8)

Original: However, the studies were based on the patients in a single place, which may be under-represented and lead to limited extrapolation.

Revised: However, the studies were based on the patients in a single place, which may not be representative of the broader Chinese population and thus lead to limited extrapolation.

Clearly state the research aim: Conclude the introduction by restating the aim of the study, which is to investigate the HRQOL and economic burden of children with pneumonia in three regions of China with different levels of economic development.

Our response: Thanks for this comment. In this revision, we have modified the statement of research aim. (Page 5, line 11)

Limitations: It is important to mention following limitations in discussion when interpreting the findings of the study and to acknowledge the potential impact they may have on the generalizability and validity of the results.

Limited generalizability: The study was conducted in three provincial capitals in China, which may not be representative of the entire country or other regions with different socioeconomic statuses. Therefore, the findings of the study may not be applicable to the broader population of children with pneumonia in China.

Selection bias: The study only included pediatric patients who were hospitalized at the time of the study, which may introduce selection bias. Children with less severe cases of pneumonia who were managed on an outpatient basis or treated at primary care clinics were not included in the study. This could limit the representativeness of the study population and affect the generalizability of the findings.

Lack of control group: The study did not include a control group of children without pneumonia for comparison. Having a control group would have allowed for a better understanding of the specific impact of pneumonia on HRQOL and economic burden, as compared to children without the disease. Without a control group, it is difficult to attribute the observed outcomes solely to pneumonia.

Self-reporting bias: The study relied on self-reporting of HRQOL by the children's parents/guardians, which introduces the possibility of reporting bias. Parents/guardians may have subjective perceptions or may not accurately assess the HRQOL of their children. Objective measures or clinical assessments could have provided a more comprehensive and accurate evaluation of HRQOL.

Lack of long-term follow-up: The study only measured HRQOL and economic burden at admission and discharge from the hospital. Long-term follow-up data would have provided valuable insights into the trajectory of HRQOL and economic burden over time and allowed for a more comprehensive assessment of the impact of pneumonia on children.

Furthermore, the study primarily focuses on the economic burden and health-related quality of life (HRQOL) of children with pneumonia, but it does not consider other potential factors that could impact these outcomes. For example, the study does not explore the impact of comorbidities or the severity of pneumonia on HRQOL and economic burden. The inclusion criteria for the study should include clear definitions or criteria for clinical diagnosis of pneumonia and no concomitant diseases. These factors could significantly influence the results and should be considered for a more comprehensive understanding of the topic.

Finally, the study uses the EQ-5D-Y instrument to measure HRQOL, which is a widely used tool. However, the study acknowledges that the utility value set for the EQ-5D-Y is not available, and therefore, it uses the Chinese value set of the EQ-5D-5L to calculate health utility scores. This introduces some uncertainty in the accuracy of the HRQOL measurements.

Our response: Thanks for this valuable comment. In this revision, we have revised the limitation section as suggested. (Page 14, line 3)

Original: First, the Chinese Y-5L value set has not been available, so the Chinese EQ-5D-5L value set was used to calculate the HUS instead. In addition, the Y-5L was designed mainly for children over 6 years of age, and the reliability and validity of its proxy version need to be further tested when it is applied to the guardians of children under 5 years. Second, although we excluded the pneumonia patients with other acute diseases, there may be undiagnosed comorbidities during data collection, which may affect HRQOL scores. Third, we did not collect certain clinical characteristics including disease severity and treatment modality, thus their influence on HRQOL and economic burden cannot be assessed.

Revised: First, the study was conducted in three areas in China, which may not be representative of the entire country or other regions with different socioeconomic statuses. Also, only pediatric patients who were hospitalized at the time of the study were included; while those who were managed on an outpatient basis or treated at primary care clinics were not included. In addition, there were missing values in the responses to the EQ-5D-Y dimensions for the patients in Shanghai. These may have limited the representativeness of the study population and affected the generalizability of our findings. Second, due to the study design and resources, we did not enroll a control group of children without pneumonia for comparison. Thus, it may be difficult to attribute the observed results exclusively to pneumonia. Third, the Y-5L was designed mainly for children over 6 years of age, and the reliability and validity of its proxy version need to be further tested when it is applied to the guardians of children under 5 years. Also, the Chinese Y-5L value set has not been available, so the Chinese EQ-5D-5L value set was used to calculate the HUS instead. This introduces some uncertainty in the accuracy of the HRQOL measurements. Fourth, we did not collect certain clinical characteristics including disease severity and treatment modality, thus their influence on HRQOL and economic burden cannot be assessed. Fifth, although we excluded the pneumonia patients with other acute diseases, there may be undiagnosed comorbidities during data collection, which may affect HRQOL scores. Last but not least, the study lacked long-term follow-up, thus it was not possible to know the trajectory of HRQOL and economic burden over time.

Responses to Reviewer#2's comments

General:

For cost, it is unlikely that OLS regression will be appropriate. First, the residuals are very unlikely to be normally distributed with constant variance (which is an assumption). Second, I think it's like that interest is not exclusively in the mean cost, but in the higher quantiles (and maybe the lower ones, as well). That is, the researchers probably would be interested in which kids have very high or very low cost. Therefore, I suggest using quantile regression, which makes no assumptions about the residuals and allows investigation of the quantiles.

Our response: Thanks for this comment. In this version, we have re-analyzed the factors influencing costs using quantile regression and have updated the methods and results.

p. 3 The fact that those 5 countries are 54% of pneumonia cases is not too surprising, since they are 46% of the world's population.

Our response: Thanks for this comment. In this version, we have deleted the sentence.

p. 5 The paragraph starting "The five response options" is very confusing. Did the parents use a VAS or a set of ordinal responses?

Our response: Thanks for this comment. Indeed, the patients also used a VAS. In this version, we have modified the sentence to make it clearer. (Page 6, line 23)

Original: The five response options of Y-5L are...Y-5L designs a visual analog scale (VAS) with a score of 100 at the top for "best imaginable health" and 0 at the bottom for "worst imaginable health".

Revised: The five response options of each dimension of Y-5L are...Y-5L also uses a visual analog scale (VAS) with a score of 100 at the top for "the best imaginable health", and 0 at the bottom for "the worst imaginable health".

p. 6 Please say how the regression models were built -- that is, how were variables selected?

Our response: Thanks for this valuable comment. Based on the evidence of previous empirical studies, the variables that may have a significant effect on HUS and costs were selected. In this version, we have added the selection basis and references in the methods section. (Page 8, line 5; Page 8, line 16)

Original : Nil.

Revised: According to previous studies (18,19), hospitalization days and demographic characteristics including region, gender, age, insurance status of the children; and education level, employment of the guardians may have a significant effect on HUS. These factors were adopted as the independent variables in the two models (hospital days was adopted only in the model for HUS difference).

Original : Nil.

Revised: Based on the evidence of prior studies (11, 12, 22, 23), gender, age, hospital days, insurance status, and disease prognosis may have a significant impact on costs and adopted as independent variables in the two models. Also, VAS score reflecting the disease prognosis, region and employment of guardians were also adopted in the models.

The fact that the *data* are not normally distributed is not relevant, OLS regression makes assumptions about the residuals, but I agree that the residuals are likely to be non-normal. However, rather than take log of costs (which is going to be kind of hard to interpret) I suggest quantile regression (see above).

Our response: Thanks for this comment. In this version, we have re-analyzed the effect of the factors using quantile regression.

p. 7 For child age it would be better to use median and IQR rather than mean and SD. After all, kids can't be less than 0 years old. (In the table, you do give the range, which helps, but IQR would also be good).

Our response: Thanks for this useful suggestion. In this version, we have used median and IQR to describe age. (Page 2, line 11; Page 8, line 27; Table 1)

p. 8 Don't just give the parameters for significant predictors, give them for all predictors (see above about how the models were built).

Our response: Thanks for this comment. In this version, we have given the parameters of all predictors. (Table 5; Table 6)

Table 2 - there should not be any blanks in this table. Also, the column %ages should add to 100, but they do not. E.g. the total for the first column and first set of values is about 74%. If the rest are missing, then that is a significant problem that needs to be dealt with (but it might not be possible to deal with it well). This problem happens in many cases. Also, assuming that very young children have "no problem" in walking about is problematic. I agree that they wouldn't be walking about, even if completely healthy, but giving them all a 0 distorts the amount of problems they are having.

I think multiple imputation could be used for the very young kids, but I'm not sure about the other missing data.

Our response: Thanks for this comment. During the data collection process, some guardians of the children did not provide HRQOL information in Shanghai. In this version, we have acknowledged it as a limitation and refined Table 2 showing the statistics of those with HRQOL information only. We have also provided the number of the patients who were younger than 18 months in the same table. Considering that those patients cannot walk around independently, "walking about" dimension may not be appropriate to describe their health. This assumption was also used in previous studies in China. Please see the reference below:

Zheng Y, Jit M, Wu JT, Yang J, Leung K, Liao Q, et al. Economic costs and health-related quality of life for hand, foot and mouth disease (HFMD) patients in China. Plos One. 2017;12(9).

Wu, Joseph T et al. "Routine Pediatric Enterovirus 71 Vaccination in China: a Cost-Effectiveness Analysis." PLoS medicine vol. 13,3 e1001975. 15 Mar. 2016, doi:10.1371/journal.pmed.1001975

Table 5 - see above.

Our response: Thanks for this comment. In this version, we have given the parameters of all predictors.

VERSION 2 – REVIEW

REVIEWER	Dr. Peter Flom Peter Flom Consulting
REVIEW RETURNED	20-Jul-2023

GENERAL COMMENTS	The authors have addressed my concerns and I now recommend publication.
---

VERSION 2 – AUTHOR RESPONSE

Not Applicable